# Improving Cooking Skills, Lifestyle Behaviors, and Clinical Outcomes for Adults at Risk for Cardiometabolic Disease: Protocol for a Randomized Teaching Kitchen Multisite Trial (TK-MT)

**DOI:** 10.3390/nu17020314

**Published:** 2025-01-16

**Authors:** Jennifer Massa, Candace Sapp, Kate Janisch, Mopelola A. Adeyemo, Auden McClure, Natalia I. Heredia, Deanna M. Hoelscher, Tannaz Moin, Shaista Malik, Wendelin Slusser, David M. Eisenberg

**Affiliations:** 1Department of Nutrition, Harvard T.H. Chan School of Public Health, Boston, MA 02115, USA; kjanisch@hsph.harvard.edu (K.J.); deisenbe@hsph.harvard.edu (D.M.E.); 2Department of Nutrition, University of Tennessee, Knoxville, TN 37996, USA; csapp1@vols.utk.edu; 3Department of Behavioral, Social and Health Education Sciences, School of Public Health, Emory University, Atlanta, GA 30322, USA; 4Department of Medicine, David Geffen School of Medicine, University of California Los Angeles, Los Angeles, CA 90095, USA; madeyemo@mednet.ucla.edu (M.A.A.); tmoin@mednet.ucla.edu (T.M.); 5Section of Obesity Medicine, Center for Digestive Health, Dartmouth Health, Lebanon, NH 03756, USA; auden.c.mcclure@hitchcock.org; 6Department of Pediatrics and Medicine, Geisel School of Medicine at Dartmouth, Hanover, NH 03755, USA; 7Department of Health Promotion & Behavioral Sciences, School of Public Health, The University of Texas Health Science Center at Houston (UTHealth Houston), Houston, TX 77030, USA; natalia.i.heredia@uth.tmc.edu (N.I.H.); deanna.m.hoelscher@uth.tmc.edu (D.M.H.); 8Michael and Susan Dell Center for Healthy Living, UTHealth School of Public Health, Austin, TX 78701, USA; 9Health Services Research, Center for the Study of Healthcare Innovation, Implementation & Policy, VA Greater Los Angeles Healthcare System, Los Angeles, CA 90073, USA; 10Susan Samueli Integrative Health Institute, Mary and Steve Wen Cardiovascular Division, Department of Medicine, University of California-Irvine, Irvine, CA 92697, USA; smalik@hs.uci.edu; 11Semel Healthy Campus Initiative Center, Chancellor’s Office, University of California at Los Angeles, Los Angeles, CA 90024, USA; wslusser@conet.ucla.edu; 12Department of Pediatrics, David Geffen School of Medicine, University of California at Los Angeles, Los Angeles, CA 90024, USA; 13Department of Community Health Sciences, Jonathan and Karin Fielding School of Public Health, University of California at Los Angeles, Los Angeles, CA 90024, USA

**Keywords:** teaching kitchen, protocol, culinary medicine, cardiometabolic abnormalities, feasibility, pilot study

## Abstract

Background/Objectives: This protocol describes a study to investigate the feasibility and preliminary efficacy of a novel Teaching Kitchen Multisite Trial (TK-MT) for adults with cardiometabolic abnormalities. The TK-MT protocol describes a hybrid lifestyle intervention combining in-person and virtual instruction in culinary skills, nutrition education, movement, and mindfulness with community support and behavior change strategies. This 18-month-long randomized controlled trial aims to evaluate the feasibility of implementing a 12-month, 24 class program, assess preliminary study efficacy, and identify barriers and facilitators to implementation. Methods: The intervention program includes 16 weeks of intensive hands-on culinary and lifestyle education classes followed by eight monthly virtual classes. Psychometric assessments and biometric data will be collected at baseline, 4, 12, and 18 months. Semi-structured interviews and open-ended surveys will be conducted during the 12-month follow-up assessment. Results: Feasibility will be assessed through recruitment, attendance, and fidelity data. Secondary outcomes will analyze changes in health behaviors, biometric data, and anthropometric measures using mixed-effects regression models. Qualitative data will undergo thematic analysis. Conclusions: As envisioned and described in detail in this manuscript, this study will inform the development and implementation of reproducible, scalable teaching kitchen interventions. The protocol described here is intended to set the stage for future investigations to evaluate evidence for the impact of teaching kitchen interventions on dietary habits, physical activity, and overall health and well-being.

## 1. Introduction

The White House Challenge to End Hunger and Build Healthy Communities [1,2] brings attention to the critical need for increasing and promoting Food Is Medicine (FIM) [3,4,5] interventions that are aimed at impacting how people eat to improve their health. FIM interventions focus on nutrition security to improve the dimensions of food security and mitigate cardiometabolic risks [4,6]. Culinary medicine is an emerging field within FIM that connects culinary arts to the medical and nutrition sciences to promote healthy dietary behaviors while empowering individuals and communities to take an active role in managing their health through food [5,7,8,9]. Culinary medicine interventions typically involve interactive cooking and nutrition education sessions that aim to improve participants’ cooking knowledge, skills, and confidence to enable sustainable and healthy lifestyle changes [7,8,10].

Previous studies have shown the efficacy of culinary medicine interventions in increasing knowledge of healthy food choices, expanding food budgets, improving food preparation self-efficacy, and enhancing healthy dietary behaviors [11,12,13,14,15,16,17]. Culinary medicine interventions teach practical knowledge and techniques related to basic nutrition, recipe modification, meal planning, knife skills, food preparation, and food storage to help make healthy eating more accessible and attainable [7,8,18]. The interactive format of culinary medicine interventions fosters engagement and empowers participants to translate knowledge learned in classes into actions applied at home [7].

There is limited evidence of the long-term causal impact of culinary medicine interventions participation on clinical outcomes such as body weight, glycemic control, and lipid profiles [11]. While some studies have shown positive associations [14,15,16,19], more robust research designs, particularly randomized controlled trials (RCTs), are needed to establish a causal link between culinary medicine interventions and sustained improvements in clinical health outcomes, both as an independent modality and in conjunction with other medical therapies. Also, there is a need for more CMIs that provide culinary education to participants in their home kitchens so that class components can be learned and practiced in real-world home environments where resources, equipment, and time may be limited. Furthermore, additional studies are needed to better understand how culinary medicine interventions can be tailored to mitigate and eliminate barriers to healthy home cooking such as food accessibility, affordability, and availability.

Teaching kitchen interventions (TKIs) are a subset of culinary medicine interventions that have been described in the literature as “learning laboratories” that seek to promote overall well-being through sustained behavior changes [20]. TKIs teach hands-on culinary and lifestyle skills to participants in physical or virtual teaching kitchens [20]. TKIs are different than other lifestyle programs that address multiple lifestyle pillars such as the Diabetes Prevention Program [21] because TKIs include both didactic and experiential culinary education in addition to lifestyle changes. This broader approach acknowledges the interconnectedness of various lifestyle factors in chronic disease prevention [20]. TKIs emphasize the need to learn (a) basic nutrition information, i.e., what to eat more of, less of, and why; (b) how to shop for and prepare healthy, delicious, easy-to-make, and affordable meals at home; (c) the importance of movement, exercise, rest, and sleep; (d) the relevance and importance of mindfulness as applied to nutrition and life; and (e) how to incorporate evidence-based behavior change strategies into everyday decisions [20,22].

The majority of culinary medicine interventions reported in the literature do not address the additional lifestyle components that are included in TKIs [23]. Movement [24,25,26], mindfulness [27,28,29,30], and behavior change [31,32,33,34,35,36] interventions have been individually evaluated and shown to positively impact risk factors of chronic disease. There is emerging evidence on the compound or multi-modal effect of implementing interventions that provide culinary medicine education alongside some lifestyle components on short-term and long-term behavioral and metabolic risk factors of chronic disease [11,37,38,39]. In addition, previous studies have identified unhealthy dietary behaviors, sedentary lifestyles, and poor sleep hygiene as modifiable risk factors for chronic disease [40]. This TKI is unique in that it (1) has been co-created by numerous clinical and academic institutions that use TKs, (2) provides culinary medicine education, creates a community support system, and mitigates lifestyle risk factors that may have a greater impact on health outcomes than traditional culinary medicine interventions that only focus on cooking and nutrition education or incorporate only some lifestyle education skills, and (3) is being tested across four sites with different populations across the U.S.

This proposed study is the first Teaching Kitchen Multisite RCT (TK-MT) to implement a robust hybrid culinary medicine and lifestyle intervention over an extended study and follow-up period in four diverse U.S. locations. The specific aims of the study described herein will be to evaluate the feasibility, preliminary efficacy, and implementation of the TK-MT for adults with cardiometabolic abnormalities in diverse settings. As envisioned, the findings from this proposed study will inform the development and implementation of a subsequent, larger TKI.

## 2. Materials and Methods

### 2.1. Study Design

The TK-MT will be an 18-month RCT that provides a hybrid teaching kitchen curriculum along with grocery provision to intervention group participants. The teaching kitchen curriculum that will be used for this proposed study was developed and piloted in previous studies [1,2]. The study objectives are to assess feasibility, preliminary efficacy, and the barriers and facilitators of implementing the TK-MT in adults at risk for cardiometabolic disease.

This Teaching Kitchen Multisite Trial (TK-MT) protocol follows the Standard Protocol Items: Recommendations for Interventional Trials (SPIRIT) [41], and this study is registered on clinicaltrials.gov (NCT05628649). This study has been approved by the institutional review board (IRB) for all study sites; Harvard School of Public Health (HSPH) is the IRB of record. Data usage agreements have been created between HSPH and the four study sites. All participants will be provided with an information sheet detailing the study procedures, potential risks, and benefits. Electronic signatures will be collected for informed consent.

### 2.2. Study Setting

Eligible study sites will have a functioning teaching kitchen [42] with facilities that can accommodate up to 30 participants for hands-on cooking activities. Sites must have a core team with the expertise to deliver the TK-MT protocol: a primary investigator (PI) to lead the site-level team; a medical doctor (MD) to confirm clinical eligibility and provide medical oversight during the study; subject matter experts (SMEs) in culinary arts, nutrition education, exercise science, mindfulness training, and health coaching to facilitate class sessions; and research staff and infrastructure to conduct the clinical trial.

The TK-MT is being conducted at four U.S. academic institutions within the Teaching Kitchen Collaborative (TKC) [43] network; Dartmouth Health (DH), a rural academic medical center serving Northern New England; the University of Texas Health Science Center Houston (UTHealth Houston) School of Public Health, an urban, public university in the southwest United States, and two urban, public universities in the western United States—University of California, Irvine (UCI), and University of California, Los Angeles (UCLA). The Harvard T.H. Chan School of Public Health (HSPH) is serving as the coordinating center.

The coordinating center will oversee the conduct of the study, compliance with IRB regulatory matters, distribution of curriculum materials, maintenance of consistent data collection, and the conduct of regular primary investigator and study team meetings. HSPH will conduct power analysis, direct participant randomization, and provide overall data management and data analysis. Data will be securely stored in password-protected REDCap (14.5.25, Vanderbilt University, Nashville, TN, USA) databases maintained by each individual study site; de-identified data will be shared with the coordinating center for network-wide analyses.

### 2.3. Study Population

#### 2.3.1. Participant Eligibility

Sites will recruit adults aged 21–70 who are at increased metabolic risk based on a classification of overweight or obesity (BMI 25–39.9 kg/m^2^) and the existence of at least one metabolic abnormality (liver, blood glucose, or lipids) identified through baseline laboratory assessments (Table 1).

Participants must have reliable Internet access at home and two devices such as a smartphone, tablet, or computer that they are comfortable operating independently (at least one must have a camera) (smartphone, tablet, or computer). Hotspots can be provided to interested participants who do not have reliable Internet access. Participants must have minimal operating cooking equipment and utilities at home, including a cooktop, oven, refrigerator, electricity, and running water.

The exclusion criteria are as follows: individuals who currently take diabetes or weight loss medications (except for metformin); have a history or current diagnosis of Type 1 or 2 diabetes; history of weight loss surgery; severe food allergies; or plan to relocate within two years. Any individuals planning to participate in other longitudinal or intensive culinary medicine or weight management programs during the study period or have participated in the past. Any individuals with recent psychiatric hospitalizations, significant mental health diagnoses, recent life-threatening illnesses, alcohol or substance misuse issues, or limitations that would prevent safe participation in the program. Individuals who are imprisoned, unable to provide informed consent, or unable to participate in all study activities. And individuals who are currently or planning to become pregnant within 18 months.

#### 2.3.2. Sample Size Estimation

A proposed sample size of 320 participants (4 study sites × 2 cohorts × 20 participants per arm) will have 80% power to detect significant improvements in metabolic syndrome biomarkers ranging from 14.2 to 30.15%, assuming that control group participants have improvements ranging from 5 to 15% for ɑ = 0.05 by using a two-sample *t* test. Power analysis calculations for sample size are based on reported effect size estimates of metabolic syndrome markers (Appendix A) [44].

#### 2.3.3. Recruitment and Randomization

Potential participants will be recruited through the TK-MT website, paid social media advertisements, physical and digital flyers, social media postings, institutional emails, television spots, ads in community newspapers, booths at health fairs, and referrals from research or clinical staff at the study sites. Study recruitment will follow the process outlined in Figure 1. Preliminary study eligibility will be assessed through an optional pre-screening survey on the TK-MT website followed by a screening phone call.

Alternatively, interested individuals can call the study team directly and complete a phone call screening. Those who are likely eligible based on the phone call screening will complete a follow-up video call with a trained research assistant (RA). These screening methods will collect eligibility data and visually confirm minimal operating kitchen equipment described in the inclusion criteria. The RA will explain the study details and obtain informed consent during these screening calls. Final eligibility will be confirmed by study site PI based on a review of anthropometric data, laboratory data, self-reported medication lists, and study site MD recommendations.

Eligible participants will be randomized at the site level in blocks to either the TK-MT intervention group or the wait-list control group receiving standard care (i.e., followed as usual by their primary care physician). Each site will independently enroll 1–2 cohorts and randomize 28–48 participants to each study arm (for single-cohort sites) or 14–30 participants per arm (for double-cohort sites); the wait-list control group will receive a condensed version of the intervention curriculum after completing the 12-month assessments. Individuals who share a household can participate in this study only if they have separate kitchen workspaces for virtual classes and must be randomized to the same study arm. All recruitment and randomization data will be documented in site-specific REDCap surveys and forms. Incentives, in the form of gift cards and small gifts (notepad, measuring spoons, etc.) valued at a total of USD 200 per individual will be provided after completing (1) laboratory and anthropometric screening, (2) baseline psychometric assessments, (3) 4-month assessments, (4) 12-month assessments, and (5) 18-month assessments.

### 2.4. Study Procedure

The intervention group will be provided with 16 weeks of intensive, hands-on culinary and lifestyle skills instruction (2 in-person and 14 virtual), followed by 6 monthly, virtual classes. Intervention and control group participants will be assessed at baseline, 4 months, 12 months, and 18 months. Control group participants will be provided with 2 in-person and 14 recorded videos of intervention classes after the 12-month assessment timepoint. See Table 2 for assessment timetable. Enrollment began in August 2023 and all participant study activities will end in October 2025.

#### 2.4.1. TK-MT Intervention

The 12-month TK-MT intervention program will be delivered by a culinary educator, a registered dietitian, and behavioral/lifestyle experts with health coaching and motivational interview training. There will be a minimum of two instructors facilitating each class; facilitator roles may change throughout intervention. The first 2 in-person classes will be held at academic or community-based teaching kitchens affiliated with each study site. For subsequent classes, participants will join via a video conferencing platform and cook in their home kitchens while facilitators lead from a teaching kitchen. The first 16 core classes (classes 1–16) will be held weekly and last approximately 2 h. The remaining 8 booster classes (classes 17–24) will be held monthly and last approximately 2 h. RAs trained in intervention implementation will be present during classes to assist participants. An overview of the TK-MT educational intervention is detailed in Table 3.

##### TK-MT Intervention Curriculum

The TK-MT curriculum is grounded in the social cognitive theory (SCT), which emphasizes the impact of behavioral capability, self-efficacy, and social influences on shaping health behaviors [45]. The curriculum combines observational learning through live modeling demonstrations with hands-on practice and skill-building activities in cooking, nutrition and food literacy, mindfulness, and physical activity. Participants observe instructors and peers, gaining cooking knowledge, skills, and self-efficacy to replicate these behaviors on their own. The TK-MT curriculum fosters self-efficacy by providing opportunities for mastery experiences. Participants receive ongoing feedback, troubleshoot challenges, and gain positive reinforcement as they learn and practice new skills. Successfully planning, preparing, and enjoying healthy meals strengthens their belief in maintaining these behaviors over time. The TK-MT curriculum also fosters a sense of community and social support. Through group discussions and activities, participants learn from and motivate each other. This social interaction builds a support network that can reinforce healthy habits beyond the program duration. By integrating these SCT principles, the TK-MT curriculum seeks to empower participants with the knowledge, self-efficacy, and social support needed to make lasting changes that contribute to a healthier lifestyle.

The TK-MT curriculum is based on the Harvard Healthy Eating Plate [46] and the Substance Abuse and Mental Health Services Administration’s (SAMHSA) Wellness Initiative [47]. The Harvard Healthy Eating Plate emphasizes the importance of consuming a high-quality, diverse diet that is rich in healthy carbohydrates, low in sugary beverages, and incorporates heart-healthy oils [46]. (An overview of the curriculum content can be found in Appendix B). The SAMHSA Wellness Initiative frames wellness as an interconnected, complex concept that functions across 8 different dimensions: emotional, physical, occupational, intellectual, financial, social, environmental, and spiritual [47]. The details of the culinary, nutrition, and lifestyle curriculum components were outlined by the TKC expert multi-day meetings (Appendix C). Mindfulness, physical activity, and behavior change (e.g., goal setting) exercises come from TKC members trained as health coaches and exercise specialists.

##### Grocery Acquisition

Sites will vary in methods used to provide study participants with their groceries to help compare and determine which practice would be the best fit for future replication. Sites may use grocery delivery to participant homes, curb-side pick-up at grocery stores, or pick-up at the study site for recipe ingredients needed for class participation. Sites will determine their method based on factors such as availability of delivery services, travel complexity, and staff availability for grocery bag assembly and distribution. Each study site will document their method of grocery acquisition along with resource requirements (food cost, labor cost, time, site level capacity).

#### 2.4.2. Usual Care (Wait-List Control Group)

Participants randomized to the wait-list control group will receive standard care (as defined by their primary care physician) for 12 months and will be offered the opportunity to participate in an abbreviated version of the TK-MT intervention following the 12-month assessment period. This version will include two in-person classes to gain practical hands-on cooking experience, educational handouts and recipes from the full 12-month TK-MT program, access to the study website, and audiovisual recordings of classes 3–16. The control group will not receive grocery ingredients to prepare the recipes in the program. At the 18-month assessment timepoint, control group participants will be sent a REDCap survey via email to assess usability, satisfaction, and overall experience of the abbreviated TK-MT intervention.

### 2.5. Outcome Measures and Data Collection

Sociodemographic data such as age, gender, race, ethnicity, income, and education, along with reasons for participation, will be collected at baseline from all participants via an online REDCap survey. Food security will also be assessed at baseline using the Hunger Vital Sign screener along with questions asking about types of food assistance received and distance to the closest grocery store. In addition, study staff will collect participants’ current medication information via a video or phone call at baseline as part of eligibility screening and enter the information into the REDCap medication form.

Self-reported medications will be reviewed during the 4-, 12-, and 18-month follow-up assessment points to assess dosage changes over the course of the study; medication changes will be treated as confounders as well as considered as independent outcomes of the teaching kitchen intervention. Given the length of this study, participants in both arms will be encouraged to receive ongoing care from their primary care physician, including medication management as appropriate.

#### 2.5.1. Primary Outcomes

The primary aim of this study will be to assess the feasibility of implementing the TK-MT. Feasibility will be evaluated across common trial-level and intervention-level outcomes [48]. Trial-level feasibility outcomes are recruitment capacity and retention. Intervention-level feasibility outcomes are participant attendance, participant engagement, intervention fidelity, intervention resources, and participant and staff acceptability. The detailed data collection methods that will be used to measure feasibility outcomes are described in Table 4.

#### 2.5.2. Secondary Outcomes

##### Preliminary Efficacy

The secondary aim of this study is to assess preliminary efficacy of the TK-MT on health outcomes. Preliminary intervention efficacy outcomes are changes in health behaviors (cooking and eating behaviors, physical activity, sleep quality, overall health, and behavior change); anthropometric measures (BMI, blood pressure, and waist circumference); and biometric measures (glycemic control, lipid metabolism, and liver function). 

Self-reported questionnaires, laboratory data (blood), and body anthropometrics (weight, height, BMI, waist circumference, blood pressure) will be collected at baseline, 4, 12, and 18 months. These data will be entered into REDCap and analyzed to assess for changes from baseline to assessment timepoints (see Table 5).

##### Behavior Changes

Cooking and Eating Behaviors

The Teaching Kitchen (TK) Core Survey is a 28-item questionnaire designed to assess self-reported eating habits, cooking habits, cooking frequency, food acquisition habits, meal planning habits, confidence related to purchasing, planning, preparing meals, and physical activity behaviors. This survey comprises modified survey items from a variety of sources including the Exercise Vital Sign (EVS) screener [49], the Cooking With Chefs (CWC) cooking confidence scale [50], and the Gallup cooking frequency questionnaire [51].

Mindful eating will be assessed using 25 of the 28 items from the mindful eating questionnaire (MEQ) [52]. The MEQ evaluates feelings of disinhibition, awareness, external cues, emotional response, and distraction related to food and eating [52]. There is strong confidence in the MEQ to assess mindful eating constructs in U.S. adults with varying BMIs [52]. Diet quality will be assessed using the 27-item Prime Diet Quality Score 30-day food frequency screener (PDQS-30D) [53]. The PDQS-30D quickly provides a diet quality score based on self-reported food consumption over the past 30 days [53]. The PDQS-30D has been assessed for concurrent validity against the Automated Self-Administered 24-Hour Dietary Assessment Tool (ASA24) [54] and the Healthy Eating Index (HEI) 2015 [55,56] metrics in adult women in the United States [53] (see Table 5).

Cooking self-efficacy will be measured using an original tool developed by the UTH team to assess participants’ perceived confidence to overcome barriers in preparing and consuming healthy meals. The self-efficacy tool has been preliminarily tested through the generation of a 40-item pool and reviewed by researchers in the nutrition and behavioral theory field. Items have been ranked for final inclusion in the self-efficacy scale (11 items). A bipolar 5-point Likert scale has been chosen to capture “how confident” respondents are to consume and prepare healthy meals despite facing barriers where 1 indicates “not at all confident” and 5 indicates “extremely confident”, with all points labeled to reduce ambiguity. Furthermore, a neutral option has been included to help understand if respondents’ self-confidence is “neutral” regarding certain statements.

Lifestyle Behaviors

Changes in physical activity will be assessed at baseline and all follow-up points using the International Physical Activity Questionnaire (IPAQ) [57]. IPAQ is a validated tool that assesses physical activity levels across various domains (e.g., work, leisure, transportation) [57]. Weekly changes in physical activity will be assessed during the weekly participation REDCap survey using 2 items from the EVS [49]. The American Psychology Association (APA) Diagnostic and Statistical Manual of Mental Disorders 5th Edition (DSM5) sleep quality questionnaire will be used to assess participants’ sleep quality and patterns, including sleep duration, sleep onset latency, and sleep efficiency [58]. It will be administered at baseline and all follow-up points to examine potential changes in sleep habits. The RAND 20-Item Short Form Health Survey (SF-20) is a well-established survey that will be administered at baseline and follow-up points to assess participants’ overall health status, including physical and mental components [59]. It will provide a comprehensive overview of participants’ health-related quality of life. The University of Rhode Island Change Assessment (URICA) measures readiness to change in various health behaviors, including diet and exercise [60]. It will be administered at baseline and post-intervention to assess changes in participants’ motivation and commitment to adopting healthier habits.

##### Laboratory Data

Fasting blood glucose, fasting insulin, hemoglobin A1c, ALT, and AST (collected as a hepatic panel), and fasting lipid profile (total cholesterol, triglycerides, HDL, and LDL) will be collected by venipuncture at baseline, 4, 12, and 18 months. Data will be collected at affiliated or at independent laboratories. All laboratory data collections will be conducted according to SOPs regardless of study site or collection location. Once lab results are received by the study team, staff will enter the data into the REDCap laboratory data form (see Table 5), which will be reviewed by the site medical director. Participants will be notified of any concerning values and given a letter to discuss with their primary provider. Study teams will assist in this communication (see Table 5).

##### Anthropometrics

Weight, height, BMI, blood pressure, and waist circumference measurements will be collected at baseline, 4, 12, and 18 months by a trained member of each study team using calibrated equipment in a lab or clinic setting following this study’s outlined standard operating procedures (SOPs). Collected measurements will be entered into the REDCap anthropometrics form by research staff (see Table 5).

**Table 5 nutrients-17-00314-t005:** Secondary Efficacy Outcomes of the Teaching Kitchen Multisite Trial.

Indicator	Assessment Tool	Description
Cooking behaviors		
Δ Cooking and eating habits	TK Core Survey	28 items assessing eating habits, cooking habits, cooking frequency, food acquisition habits, confidence r/t food acquisition, meal planning, and cooking
Δ Cooking skills self-efficacy	Cooking Self-Efficacy Survey	11 items assessing cooking confidence and self-efficacy
Lifestyle behaviors		
Δ Diet intake	PDQS-30D [53]	27 items assessing foods/beverages consumed over the past month
Δ Mindful eating	MEQ [52]	28 items assessing mindful eating
Δ Physical activity	IPAQ [57]	7 items assessing physical activity type, frequency, and duration
Physical activity frequency Physical activity duration	EVS	Weekly assessment of changes in physical activity type, frequency, and duration
Δ Sleep habits	PROMIS APA DSM5 [58]	8 items assessing sleep quality
Δ Physical health	URICA [60]	12 items assessing readiness to change r/t physical health
Δ General health	CDC Healthy Days Measures [61]	5 items assessing physical health, mental health, overall health, and smoking habits
Δ Quality of Life	RAND 20 SF [59]	20-Item Short Form Health, quality of life, quality of work–life balance survey
Laboratory Measures		
Δ Glycemic function	Fasting serum glucose [62,63]	Measure of glucose circulating in serum while in a fasted state
	Fasting serum insulin [64]	Measure of insulin circulating in blood while in a fasted state
	Hemoglobin A1C	Estimate of average glucose concentration in blood over the past 120 days
Δ Liver function	Aspartate transaminase (AST) Alanine transaminase (ALT)	Markers of hepatocellular injury
Δ Lipid profile	Total cholesterol High-density lipoprotein Low-density lipoprotein Triglycerides	
Anthropometric Measures		
ΔBMI	BMI = weight (kg)/height (m)^2^	Change in anthropometric measures at 4, 12, and 18 months
Δ Blood pressure	Electric blood pressure monitors	
Δ Waist circumference	Measuring tape performed by trained staff	

Δ = change in.

### 2.6. Intervention Implementation

Another secondary aim will be to identify barriers and facilitators of implementation. Qualitative assessments will occur though open-ended questions on staff and participant feedback surveys and through semi-structured interviews with intervention participants. Participant interviews will be conducted at the 12-month assessment timepoint. Interviews will assess acceptability, usefulness, strengths, weaknesses, compliance, and missing elements of this study, intervention curriculum, and intervention delivery. A trained study team member will conduct and record audio of each interview for transcription. Interviews will take place via phone, video call (voice only), or in-person. All participant data will be de-identified. Open-ended survey questions will also be included in participant satisfaction surveys.

Open-ended staff surveys will evaluate barriers and facilitators to meeting participant needs, perceived need for innovation, and participant feedback.

### 2.7. Data Analysis

The primary outcome of feasibility will be assessed through frequency data that are collected on recruitment, attendance, and fidelity. Descriptive statistics (proportions for rates, means, or medians for satisfaction scores) will be used to summarize these measures. We will also document reasons for non-completion (withdrawal or attrition). For surveys addressing participant satisfaction, descriptive statistics (means, medians, proportions, 95% confidence intervals [CI]) will be computed, and data will be analyzed to assess for change over the 16-week intensive curriculum, 8-month boosters, and 18-month sustainability measures. A paired *t*-test will evaluate pre/post changes in continuous measures. 

Secondary associations considering participation in the TK-MT and the biometric data (change in body weight and clinically relevant change in one of the following metabolic markers: fasting plasma concentrations of glucose, insulin, triglyceride, cholesterol, HDL, LDL, and liver function as AST and ALT) will be analyzed using a linear mixed-effects regression model for continuous outcomes. 

Descriptive statistics (means, medians, proportions) will be used to summarize baseline characteristics and changes in these outcomes (self-reported health behaviors, lab values, and anthropometrics) over time. Baseline and demographic data will be controlled for in this model. Lastly, we will investigate whether missing data, due to dropouts and poor attendance, are non-ignorable through a sensitivity analysis based on the imputation approach developed by co-investigator Dr. Wang and colleagues [65]. Preliminary investigation of associations and correlations will be explored graphically and formally using parametric and nonparametric tests, as appropriate. Regression diagnostics will be performed to assess the validity of the multivariate models and their assumptions, and first-order interaction terms will be included to assess potential effect modification. 

For participant interviews and open-ended survey items, thematic analysis will be conducted by the HSPH team to independently read transcripts and conduct open and then focused coding using inductive and deductive coding approaches in NVivo (15, Lumivero, Denver, CO, USA) [66].

## 3. Discussion

This protocol provides a detailed description of a multisite feasibility study to evaluate a novel hybrid teaching kitchen intervention aimed at improving health behaviors and outcomes. By combining in-person and virtual instruction, providing grocery ingredients, and integrating hands-on culinary skill development, nutrition education, physical activity promotion, mindfulness activities, and behavior change strategies, this protocol, and the study it is connected to, sets the stage for future investigations of a range of teaching kitchen and other Food Is Medicine and Whole Person Health educational interventions. 

The importance of this study lies in its potential to inform the development and implementation of effective and scalable interventions to address the growing incidence of chronic disease. 

Limitations of this study include its duration, intensity, and eligibility criteria, which could make recruitment challenging. Adults who meet the BMI and lab value criteria but are not yet on medications are a smaller subset of the U.S. population to identify and recruit. The time commitment is anticipated to limit the number of people who will be interested in this study; while there has been thoughtful consideration around the time the classes are scheduled (after work evening hours), it is unlikely to be practical for some. Access to groceries may be challenging at individual study sites. This and the location of grocery pick-up may be a limiting factor for some potential participants. Participants must also have two Internet-capable devices and be capable of operating them independently. This may limit participation. Considerations and adaptations have been made to the curriculum itself to account for regional differences in foods and flavor profiles; however, the curriculum is only being offered in English, which limits implementation and evaluation of this protocol to English-speaking populations only. 

By evaluating the feasibility and acceptability of a hybrid teaching kitchen model, this study aims to inform strategies for implementing similar interventions across diverse populations while addressing barriers to participation. In addition, the assessment of lifestyle behavior changes and biometric data will offer valuable contributions to the burgeoning Food as Medicine movement and the field of integrative medicine, emphasizing the potential of culinary education to promote health and well-being. Additionally, this study supports the growing call to enhance nutrition education in medical schools and to reintroduce home economics in elementary schools as foundational tools for lifelong health literacy. Notably, this multisite teaching kitchen intervention is the first to operationalize the dual assessment of pathogenesis and salutogenesis, as articulated by Helene Langevin and the NIH, aligning with the framework presented in her 1 November 2024 NCCIH talk in Washington, DC.

This protocol description highlights the value of sharing a robust and potentially transformative multi-institutional effort to investigate teaching-kitchen-related curricula and other Whole Person Health and Lifestyle Medicine educational interventions. By detailing the study design, methodology, and planned assessments, this paper underscores the importance of creating a replicable framework for evaluating such interventions. This approach not only contributes to the growing evidence base for innovative health education strategies but also provides a foundation for future studies seeking to explore the effectiveness of integrative and lifestyle-focused models. Sharing this protocol supports transparency, fosters collaboration, and encourages adaptation of these interventions to diverse populations and settings, ultimately advancing the fields of health promotion and preventive care.

If successful, this intervention has the potential to produce significant public health impact by improving dietary habits, increasing physical activity, and enhancing overall health and well-being. By addressing multiple health behaviors simultaneously, this comprehensive approach may lead to greater and more sustained improvements in health outcomes compared to single-component interventions.

## Figures and Tables

**Figure 1 nutrients-17-00314-f001:**
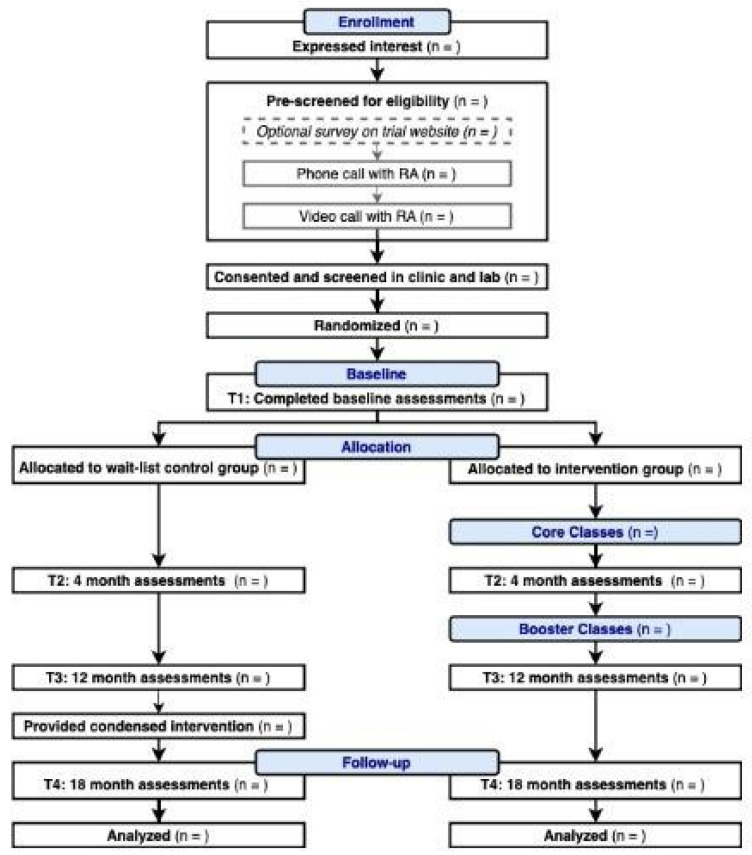
Teaching Kitchen Multisite Trial (TK-MT) Study Flowchart.

**Table 1 nutrients-17-00314-t001:** Biometric Inclusion Values for Study Eligibility (must have at least one of the following abnormal values).

Biometric Marker	Minimum Level for Eligibility	Maximum Level for Eligibility
Fasting Glucose	100 mg/DL	125 mg/DL
Hemoglobin A1C	5.70%	6.40%
Triglycerides	150 mg/DL	No upper limit
HDL	No lower limit	Men: <40; Women: <50
LDL	100 mg/DL	No upper limit
ALT	Men: >55 unit/L; Women: >30 unit/L	No upper limit

Participants must have at least 1 eligible biometric marker in addition to 25 ≤ BMI < 40 to meet inclusion criteria for this study.

**Table 2 nutrients-17-00314-t002:** TK-MT Study Assessments.

Timepoint	Evaluation Measures	Intervention Group	Control Group
Baseline	Anthropometric measurements	At study site with RA	At study site with RA
	Biometric measurements	At designated lab facility	At designated lab facility
	Psychometric assessments	Email link to REDCap surveys	Email link to REDCap surveys
Core Phase (Class 1–16)	Weekly participant satisfaction	Email link to REDCap surveys	N/a
	Weekly participant photo upload	Email link to REDCap surveys	N/a
	Participant worksheet uploads	Email link to REDCap surveys	N/a
Week 16 (4-month assessment)	Anthropometric measurements	At study site with RA	At study site with RA
	Biometric measurements	At designated lab facility	At designated lab facility
	Psychometric assessments	Email link to REDCap surveys	Email link to REDCap surveys
Booster Phase (Class 17–24)	Monthly participant satisfaction	Email link to REDCap surveys	N/a
	Monthly participant photo upload	Email link to REDCap surveys	N/a
	Participant worksheet uploads	Email link to REDCap surveys	N/a
12-month assessment *	Anthropometric measurements	At study site with RA	At study site with RA
	Biometric measurements	At designated lab facility	At designated lab facility
	Psychometric assessments	Email link to REDCap surveys	Email link to REDCap surveys
	Program feedback surveys	Email link to REDCap surveys	N/a
	Interviews (participants)	Email login info for interview	N/a
18-month assessment	Anthropometric measurements	At study site with RA	At study site with RA
	Biometric measurements	At designated lab facility	At designated lab facility
	Psychometric assessments	Email link to REDCap surveys	Email link to REDCap surveys
	Program feedback	N/a	Email link to REDCap surveys

*. Program feedback survey will be completed by participants and research staff at the 12-month assessment point.

**Table 3 nutrients-17-00314-t003:** TK-MT Intervention Overview.

Class Type	Cooking and lifestyle instruction by chef educators, registered dietitians, and lifestyle experts. ○ In-person: classes 1 and 2. ○ Virtual: classes 3–24.		
Class Duration	12-month program. ○ 2 h/session. ▪ 16 weekly classes (core classes). ▪ 8 monthly classes (booster classes).		
Class Structure	Introduction and discussion of class objectives. Didactic education on selected nutrition and lifestyle components with group discussion or worksheet opportunities. Discussion of recipe and ingredients. Cooking of recipe. Mindful exercise, meditation, or physical activity. Taste and eat prepared meal and discuss shared experiences. Time for open discussion, reflection, and/or troubleshooting.		
Behavioral Framework	Observational/ hands-on learning. Social influence.	Self-efficacy. Knowledge building. Attitudes and beliefs.	Peer support. Skill development and practice.
Wellness Domains	Social. Emotional. Spiritual.	Intellectual. Physical. Environmental.	Financial. Occupational.

**Table 4 nutrients-17-00314-t004:** Primary Feasibility Outcomes of the Teaching Kitchen Multisite Trial.

Indicator	Quantitative Assessment	Qualitative Assessment
Recruitment Rate	$\left(\frac{\#randomized}{\#screened}\right)\times 100\%$ REDCap randomization module. REDCap screening survey (completion status).	Barriers and facilitators to recruitment based on: Notes from each site’s research team. Feedback from individuals who choose not to participate. Participant feedback during post-study interviews.
Retention rate	$\left(\frac{\#attended>80\%\;classes}{\#randomized}\right)\times 100\%$ REDCap class participation survey (attendance item). REDCap randomization module.	Barriers and facilitators to retention based on: Notes from each site’s research team. Feedback from participants who choose to withdraw from the study.
Attendance rate (per participant)	$\left(\frac{\#classes\;attended}{\#total\;classes}\right)\times 100\%$ REDCap class participation survey (attendance item).	Barriers and facilitators to attendance based on: Notes from each site’s research team. Self-reported free text response to why a class was missed in REDCap class participation survey. Feedback from participants during post-study interviews.
Participant engagement	Post-class recipe photo upload rate = $\;\;\frac{\Sigma \left(\frac{\#recipe\;photos\;uploaded}{\#total\;recipes}\right)}{\#intervention\;participants}\times 100\%$ Post-class worksheet upload rate = $\;\;\frac{\Sigma \left(\frac{\#worksheets\;\;uploaded}{\#total\;worksheets}\right)}{\#intervention\;participants}\times 100\%$	Barriers and facilitators to participant engagement based on: Notes from each site’s research team. Feedback from participants during post-study interview.
Intervention fidelity	Session fidelity = $\bar{fs_{i}}$ , where *fs_i_* = post-class fidelity checklist (possible score of 0–14 points).	Barriers and facilitators to intervention fidelity based on: Free text response to post-class fidelity checklist items. Feedback from staff in the end-of-study staff surveys.
Intervention resources	Planned intervention costs vs. actual, grocery, labor, and overhead costs.	
Average class satisfaction (cs) rate Average class recipe satisfaction (rs) rate	REDCap class satisfaction survey = $\frac{\Sigma \bar{cs_{i}}}{\#respondents}\times 100\%$ REDCap recipe satisfaction survey = $\frac{\Sigma \bar{rs_{i}}}{\#respondents}\times 100\%$	Barriers and facilitators to individual session acceptability based on: Notes from each site’s research team in the REDCap implementation form. Free text response to perception of the class, recipe, and grocery experience following each class session in the REDCap class participation survey. Feedback from participants during post-study interviews.
Overall program acceptability	Program satisfaction (5-pt Likert scale). Recipe ease (5-pt Likert scale). Recipe preparation success (5-pt Likert scale). Intent to continue using recipes (yes/no). Top 3 recipes. Perceived cultural appropriateness (yes/no). Most helpful program component. Skills learned. Food access. Cooking equipment access. Likelihood to recommend the program. Surprising and inspiring program components.	Barriers and facilitators to overall program acceptability based on: Free text responses to the end of intervention survey. Feedback from participants during post-study interviews.

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
