# Peer review of "Improving Cooking Skills, Lifestyle Behaviors, and Clinical Outcomes for Adults at Risk for Cardiometabolic Disease: Protocol for a Randomized Teaching Kitchen Multisite Trial (TK-MT)"

_nutrients, 2025, doi:10.3390/nu17020314_

Round 1
Reviewer 1 Report
Comments and Suggestions for Authors
This is an interesting protocol for a study. It is very well written. I only have 2 comments.
Table 1. For HDL should the maximum be “no lower limit” rather than “no upper limit.”
In the next paragraph you repeat the words “will be excluded” several times. Perhaps start the paragraph by saying “Exclusion criteria are as follows".
Author Response
Comments 1: Table 1. For HDL should the maximum be “no lower limit” rather than “no upper limit.”
Response 1: Thank you for pointing this out. We agree with this comment. Therefore, we have changed the text for HDL in the column labeled “minimum level for eligibility” to “no lower limit” and changed the text in the column labeled “maximum level for eligibility” to “Men: < 40; Women: < 50” which can be found on page 4, Table 1, starting at line 169 of the revised manuscript.
Comments 2: In the next paragraph you repeat the words “will be excluded” several times. Perhaps start the paragraph by saying “Exclusion criteria are as follows".
Response 2: Thank you for pointing this out. We agree with this comment. Therefore, we have added “the exclusion criteria are as follows” at the beginning of the paragraph on page 4, paragraph 3, line 172 of the revised manuscript. We have removed the redundant “will be excluded” phrases on page 4, paragraph 3, lines 172-182 of the revised manuscript.
Reviewer 2 Report
Comments and Suggestions for Authors
The article titled presents a well-prepared research protocol that enables the evaluation of the effectiveness of a program combining culinary education, healthy habits, and social support. The authors clearly described how they plan to assess the feasibility and effectiveness of such an intervention, demonstrating that the project is well thought out.
The intervention includes both practical and virtual sessions, and various methods, such as health measurements and qualitative analyses, will be used to evaluate the outcomes. This comprehensive approach allows for a better understanding of the program’s impact on participants’ health.
However, the tables included in the protocol, especially the one labeled Table 6 and Appendix 2, are difficult to read and need better formatting. Simplifying and presenting them more clearly would make the data easier to interpret. Additionally, there is a numbering error in the tables—after Table 4, the next table is labeled as Table 6. This issue should be corrected to avoid confusion.
There is also a lack of a more detailed discussion about potential challenges/limitations that might arise during the project. Such information could be valuable for those who aim to implement similar programs in the future.
Author Response
Comments 1: The tables included in the protocol, especially the one labeled Table 6 and Appendix 2, are difficult to read and need better formatting. Simplifying and presenting them more clearly would make the data easier to interpret.
Response 1: Thank you for pointing this out. We agree with this comment. Therefore, we have reformatted all tables by simplifying the content and removing excessive spacing so they are easier to read and interpret. Appendix B has been split into two tables, 1 table to describe the intensive classes and another table to describe the booster classes.
Comments 2: Additionally, there is a numbering error in the tables—after Table 4, the next table is labeled as Table 6. This issue should be corrected to avoid confusion.
Response 2: Thank you for pointing this out. We agree with this comment. Therefore, we have changed the title of Table 6 to Table 5. We have also corrected the in-text references to the table to reflect the updated table numbering.
Comments 3: There is also a lack of a more detailed discussion about potential challenges/limitations that might arise during the project. Such information could be valuable for those who aim to implement similar programs in the future.
Response 3: Thank you for pointing this out. We agree with this comment. Therefore, we have added text describing limitations to recruitment, class logistics, grocery acquisition, virtual class logistics, and cultural adaptations to the discussion section starting on page 12, line 460.